# Effectiveness of Lubricants and Fly Ash Additive on Surface Damage Resistance under ASTM Standard Operating Conditions

Tuan-Anh Bui [1],* , Van-Hung Pham [1], Duc-Toan Nguyen [1],* and Ngoc-Tam Bui [2]

[1]  School of Mechanical Engineering, Hanoi University of Science and Technology, Hanoi 10000, Vietnam
[2]  Department of Machinery and Control Systems, Shibaura Institute of Technology, Tokyo 135-8548, Japan
*   Correspondence: anh.buituan@hust.edu.vn (T.-A.B.); toan.nguyenduc@hust.edu.vn (D.-T.N.)

**Abstract:** This study investigated the effectiveness of lubricants and additives in preventing surface damage and wear, which is critical for numerous industrial applications. The ASTM standard operation conditions were employed for a series of experiments using a four-ball friction and wear tester, testing three different oils (A, B, and C) with and without 0.5% fly ash additive. The experiments were analyzed using a microscope to evaluate the effectiveness of oils and additives in preventing surface damage. The study found that certain anti-wear additives significantly reduced the size of wear scars on the balls, indicating their effectiveness in reducing surface damage. These findings have important implications for developing new lubricant formulations and optimizing industrial processes that involve sliding and rolling contacts. The study emphasizes the importance of selecting appropriate oils and additives for specific applications to minimize surface damage and wear, which is crucial for improving the performance and lifespan of machine components.

**Keywords:** lubricants; additives; wear resistance; four-ball friction tester; anti-wear additives

## 1. Introduction

Lubricating oil is a critical component in reducing friction and wear between machine parts, providing cooling and corrosion protection to friction surfaces, and enhancing machine efficiency, longevity, and reliability [1]. The ability of lubricating oil to withstand various factors, such as load, speed, temperature, oil aging rate, and external environment during operation, is essential for its lubricating ability and service life. The demand for industrial lubricants is high in many countries, with a typical additive ratio of 0.5%–1% [2]. The viscosity and viscosity index (VI) are crucial quality criteria for lubricants, measuring their thermal stability. To enhance the load-carrying and VI properties of lubricants, manufacturers have introduced various additive packages.

Recent studies have investigated the effect of using nano $Al_2O_3$ additives in lubricants [1,2]. The addition of aluminum oxide nanoparticles to commercial lubricating oil, such as BP Vistra, is shown to improve its lubricating properties and heat resistance. The small size of the nano aluminum oxide particles increases the layer separation between contact surfaces, reducing friction and lubricant consumption, and extending the life of the oil. Additionally, nano aluminum oxide reduces the viscosity of the oil at high temperatures, enhancing its heat resistance.

Researchers have studied the rheological properties, particularly the dynamic viscosity, of the mixture of nano aluminum oxide and base oil at different additive ratios and temperatures. The results showed that the addition of nano aluminum oxide modified the dynamic viscosity of the oil, and identified the appropriate ratios and temperatures to ensure good lubricating properties. The effect of the mixture of lubricant oil and nano aluminum oxide $Al_2O_3$ on hydrodynamic pressure and the load-carrying capacity of the bearing was also studied. The addition of nano $Al_2O_3$ created a significant change in

the hydrodynamic pressure curve around the circumference of the bearing, leading to a considerable increase in the load-carrying capacity of the bearing [2].

Numerous studies investigated the impact of small-sized particulate additives on the quality of lubricating oil, including metal oxide nanoparticles, such as $CuO$, $Fe_2O_3$, $NiO$, and $Al_2O_3$, added to lubricating oil, and the influence of temperature on the viscosity of the oil. The results of these studies provide a basis for developing suitable lubricants for desired working conditions of equipment, maintaining stable working conditions and extending the life of the equipment [3]. In a recent study, the influence of $CuO$, $Fe_2O_3$, and $NiO$ nanoparticles, and temperature on the viscosity of industrial oil was investigated. The authors conducted experiments with different ratios of nanometal oxide and varying temperatures ranging from 27–710 °C, with oil viscosity of 77 and 350 PaS. The results showed that these factors affected the viscosity of the oil mixture, leading to a 50%–70% reduction in viscosity depending on the ratio of nanometal oxide used [4]. Overall, the use of nano additives in lubricating oil shows promise for improving its properties and extending the life of equipment.

The application of nanotechnology to improve the tribological properties of lubricants gained increased interest in recent years. Various studies demonstrated that the addition of nanoparticles to lubricants can lead to a reduction in the coefficient of friction and wear rate, resulting in improved tribological performance [5–12]. Examples include the reduction in viscosity of lubricants depending on the density and size of micro- or nano-metal particles [5]; the micro-bearing effect at the friction interface of $SiO_2$ nanoparticles, leading to friction reduction [6]; and the development of eco-friendly oils with improved lubricity through adding $SiO_2$ nanoparticles to epoxidized Madhuca indica oil, resulting in a reduction in the coefficient of friction and wear rate [7]. Wu et al. [8] demonstrated that the use of hard-shelled soft-core composite nanoparticles ($ZnO@SiO_2$) as a new additive in grease led to a significant reduction in the coefficient of friction and wear scar diameter when added at 1 wt.%. Arumugam et al. [9] compared the tribological properties of chemically modified rape-seed oil with two different anti-wear nano additives—$TiO_2$ and $Al_2O_3$—and found that the oil with spherical $TiO_2$ nanoparticles had a smoother worn pin surface and a lower friction coefficient. Ionescu et al. [10] studied the effectiveness of adding $ZnO$ to rape-seed oil and found that the wear rate decreased with 1%wt $ZnO$, while the authors developed a new lubricant using dispersant at a 1:1 ratio to the additive. The impact of $CaCO_3$ and $SiO_2$ nanoparticles on lithium-based grease was investigated by Razavi et al. [11], and the authors found that the nanogreases prepared with $CaCO_3$ and $SiO_2$ nanoadditives showed improved physical, tribological, and rheological properties. Bakalova et al. [12] used nanoparticles of $TiO_2$ or $SiO_2$ in a cooling lubricant emulsion to reduce friction coefficient, wear of the cutting tool, and variance of measured values. $CuO$ and $TiO_2$ nanoparticles were added to a metal-working polymeric lubricant by Ghobadian et al. [13]: the authors observed overall improvements in tribological properties. Akl et al. [14] studied the tribological properties of $CuO$ nanoparticles in lubricating oil, observing a reduction in wear rate and friction coefficient compared to the base oil when the $CuO$ concentration was 0.75 wt.%. These studies collectively demonstrate the potential of nanotechnology to enhance the performance of lubricants in various industrial applications.

Numerous studies have shown that metal oxide nanoparticles have a decreasing effect on viscosity; these oxides result from corrosion, naturally falling into the lubrication area [1–3]. Additionally, other studies have described the effect of nanoadditives on certain lubricating properties of industrial oil [15–22]. These studies have demonstrated that metal oxide additives can improve the lubricating properties of oil and increase the life of machine parts. However, the load-carrying capacity of the oil due to the influence of these additives has not been fully addressed, making it an important task to investigate the load-carrying ability of oil mixed with fly ash additives to find a solution for stabilizing the working conditions of machine parts and changing the viscosity. This research gap is particularly relevant for evaluating the lubrication quality of heavy-duty equipment operating in harsh plant environments.

Fly ash is a super fine spherical particle comprised of silica oxide and other metal oxide particles ranging in size from nano to micro, which is obtained from the dust generated through coal-fired furnaces in thermal power plants. Therefore, a proposed solution for enhancing the load capacity of industrial lubricating oil involves adding a nano- or micro-modifying additive sourced from fly ash from thermal power plants. The fly ash additive is to be added at a concentration of approximately 0.5%–1% of the lubricating oil, creating an industrial lubricant mixture with nano/micro elements. The appropriate concentration of the additive will be studied and developed to increase the viscosity, thermal conductivity, and life of the lubricating oil mixture, reduce oil residue emissions, and contribute to sustainable development and the protection of the environment from the impacts of oil refining and lubricant waste.

This study aimed to investigate the effect of fly ash additive on the wear resistance of industrial oils, using the ASTM D4172 standard and a four-ball friction testing machine. The lubricating oil was prepared with a pre-designed plan, and fly ash additive was added at a ratio of 0.5%. Scratch measurements were taken on the steel balls, and the mass of the balls was measured before and after the experiments. The results showed that the balls with the fly ash additive had better wear resistance than those without the additive, and the level of dispersion of the mass values in cases with the additive was smaller. The ASTM D-4172 test standard is suitable as a basis for improving oil quality in research on adding additives from fly ash.

## 2. Materials and Methods

The friction properties of the lubricating oil were evaluated following the ASTM D4172 standard. Experimental equipment and test parameters were prepared and planned accordingly. The stability of the experimental equipment was checked before conducting the experiment. The main parameters that were adjusted include maintaining the oil temperature at a stable level of 75 °C with a permissible error of ±2 °C, setting the rotational speed of the upper ball to 1200 rpm with a permissible error of ±60 rpm, determining a test duration of 60 min with a permissible error of ±1 min, and applying a load of 392 N on the ball with a permissible error of ±2 N. The four-ball friction testing machine's schematic diagram is described in Figure 1. The lubricating oil was prepared according to a pre-designed plan, which involved using three types of commonly used industrial lubricating oils to evaluate some of their lubrication properties. To compare the lubrication properties of the oil mixture, the fly ash additive obtained from a thermal power plant was added to the three oils at a ratio of 0.5%.

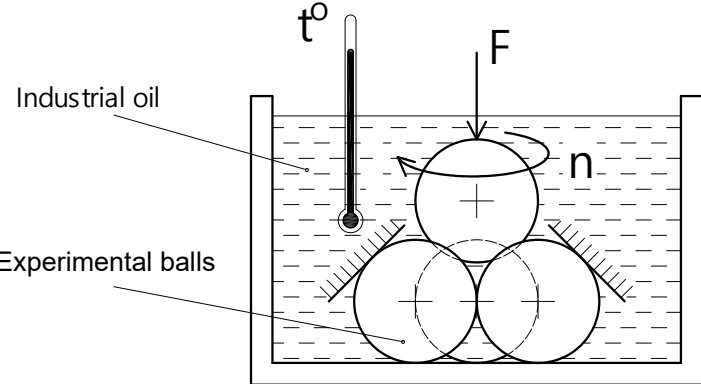

**Figure 1.** Schematic representation of four-ball machine operation.

For each test, four balls made of hard steel material with a diameter of 12.7 mm were used. Three of these balls were placed horizontally, in contact with each other, and remained fixed throughout the testing process. The fourth ball was positioned on top and in contact with all three lower balls. These balls were immersed in the experimental oil during the test.

The test was conducted through heating the oil and ensuring the temperature remained stable using a temperature sensor system. A load was applied to the top ball in a direction that pressed against the lower balls, and a rotational motion was induced in the top ball perpendicular to the plane formed of the three lower balls. The test duration was set for 60 min. Following the test, the balls were cleaned, and the size of the scratches on the contact surface of the lower balls was measured to assess the overall lubricating performance of the oil.

### 2.1. Experimental Procedure

Step 1: Clean four balls thoroughly, including the clamping parts for the upper and lower balls, and the oil cup. Use jet gasoline as a solvent to clean the balls and dry them. Use industrial lint-free wipes to clean and replace all parts, ensuring that no solvent residue is left when replacing the test oi;

Step 2: Attach one of the clean balls to the main shaft of the machine and check its tightness;

Step 3: Place three cleaned test balls in the test oil cup and secure the mechanism holding the three lower balls;

Step 4: Pour the oil to be evaluated into the test oil cup until it is at least 3 mm above the top of the balls. Ensure the oil level remains the same after the oil fills all gaps in the test oil cup;

Step 5: Install the test oil cup into the machine, and gradually apply a test load of 392 N to avoid shock loading;

Step 6: Activate the heating element and set the temperature control to heat the oil to 75 °C. The heating element will automatically turn off when the oil temperature exceeds 75 °C and vice versa;

Step 7: Start the motor to transmit rotational motion to the upper ball shaft when the temperature reaches 75 ± 2 °C. The control unit will automatically turn on the motor and run it for 60 min before shutting it off using a time relay;

Step 8: After 60 min of operation, the machine will stop, and the test oil cup and three balls will be removed;

Step 9: Use a high-precision device to measure the wear scar on the balls in the following order: drain the test oil, remove the three balls from the test cup, and clean them before taking the necessary measurements. Measure the wear scar diameter under a microscope and calculate the average after three experiments. The lubricating oil's properties can be determined from this result. Repeat this experimental procedure for each test using different oils.

### 2.2. Experimental Balls

To obtain a sufficient sample size for the planned experiments, a suitable number of balls (Figure 2) were cleaned and checked for size and surface hardness before installation in the experimental apparatus. After each experiment, the balls were thoroughly cleaned, marked, and classified to determine the required parameters.

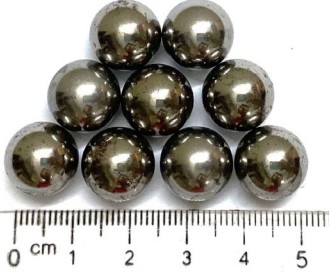

**Figure 2.** Preparation of Balls for Experiments.

### 2.3. Experimental Oils

For the experiments, industrial oils with the technical characteristics outlined in Table 1 were selected. In order to maintain the reputation of commercial industrial oils, three types of oils were assigned the names A, B, and C.

**Table 1.** Key specifications of experimental oils A, B, and C.

| Specification | Test Method | Unit | Oil A | Oil B | Oil C |
|---|---|---|---|---|---|
| Specific gravity at 15 °C | ASTM D4052 | g/mL | 0.852 | 0.868 | 0.890 |
| Kinematic viscosity at 100 °C | ASTM D445 | mm$^2$/s | 11.9 | 14.6 | 18.1 |
| Kinematic viscosity at 40 °C | ASTM D445 | mm$^2$/s | 65 | 126 | 165 |
| Viscosity index | ASTM D2270 | - | 183 | 117 | 120 |
| Pour point | ASTM D97 | °C | −45 | - | −24 |
| Cleveland flash point | ASTM D92 | °C | 234 | 205 | 226 |
| Sulfated ash | ASTM D874 | % | 1.2 | 0.88 | 0.9 |

When equipment operates at low temperatures, the oil can become thickened, resulting in difficulty starting the engine due to frictional contact without proper lubrication. Researchers added special additive compounds to reduce the viscosity dependence of lubricating oil on ambient temperature. This multi-grade oil helps the engine to start in low temperatures, while ensuring proper lubrication. The starting process is only a short stage in the engine's overall working time. When the engine operates in a stable stage with a temperature range of 40–100 °C, the viscosity dependence on temperature is considered a linear relationship. Therefore, the relationship between viscosity and temperature of multi-grade oil can be determined as a first-order relationship.

Viscosity is a significant parameter that affects the quality of lubrication, as demonstrated via the scratch test in ASTM D472 standard. However, evaluating the lubrication characteristics solely based on the viscosity index may be misleading, as several studies have shown the influence of additives on the lubrication properties of oil. This influence is reflected in the specifications provided by oil manufacturers.

The operating conditions of machine parts follow three stages: the running-in stage, the stable stage, and the failure stage. In the running-in stage, wear rate is high at the beginning of operation, but decreases over time due to the initial surface containing many peaks and valleys resulting from the machining process. As working time increases, the process of flattening out the peaks and valleys occurs more frequently, increasing the contact area between surfaces, and reducing contact pressure and wear rate. The stable stage is when the peaks and valleys have been flattened out to a significant extent, resulting in a large contact area, low contact pressure, and a significant reduction in wear rate. The failure stage occurs when the wear rate increases, and the size of the part decreases, leading to increased working stress until it reaches the allowable stress value. Usually, to ensure safety, the machine must be stopped before the working stress reaches the allowable value.

### 2.4. Fly Ash Additive

Fly ash from thermal power plants, with its spherical structure and particle size of about a micrometer, can be used as an inert additive to increase the viscosity, thermal conductivity, and service life of lubricant oil mixtures, and reduce the deposition of sludge in machinery. In this study, fly ash was selected as an additive at a concentration of 0.5% in three types of industrial oils, resulting in a mixture of oxide particles and oil. The appropriate concentration of the additive was studied to develop increased viscosity, thermal conductivity, and service life of lubricant oil mixtures, as well as reduce oil sludge emissions, contributing to green development and environmental protection from the impacts of oil refining and lubricant waste. The fly ash used in this study was obtained from a thermal power plant and contained main components (Table 2), such as $SiO_2$, $Al_2O_3$, $K_2O$, and $Fe_2O_3$ in a spherical structure, with particle sizes ranging from 50 nm to less than 1 μm.

**Table 2.** Components of fly ash additive.

| Component | Percentage | Particle Size |
|---|---|---|
| $SiO_2$ | 57.02% | |
| $Al_2O_3$ | 23.82% | |
| $K_2O$ | 6.56% | 50 nm–1 μm |
| $Fe_2O_3$ | 4.69% | |

## 3. Results and Discussion

The experiments involved the measurement of scratch sizes on the balls using a microscope measuring device. The scratch diameter of each ball in each experiment was recorded for data processing and analysis. Each experiment was repeated three times, and the ball parameters before the experiment were the average value of the selected balls for each experiment. The measurement results after the experiment were also the average value of the samples measured. Three types of industrial oils and two cases of 0.5% fly ash additive and no additive were tested. Scratch sizes on the upper and lower balls for three types of oils—A, B, and C—were observed and measured to evaluate the effect of the additive on the material's wear resistance compared to the case without the additive. Figure 3a–c shows the scratch sizes on the upper and lower balls for oils A, B, and C, respectively. These figures provide a visual representation of the scratch sizes on the balls and help in the comparative analysis of the different experimental conditions.

The friction characteristics under lubricated conditions with several types of industrial oils were evaluated using the average mass and standard deviation of the balls before and after the experiments, as listed in Table 3 and Figure 4. After a certain working time, the balls will erode, and this erosion is reflected in a decrease in the mass of the balls. In both cases, with or without additives in the lubricating oil, the balls decreased by about 0.001–0.002 g in weight. However, for each type of industrial oil used, the mass distribution is different. Most experiments showed that the average mass reduction value of the balls was 0.001 g, except for the experimental cases with oil A with or without additional fly ash additives, where the average mass of each ball decreased by 0.002 g. However, it can be seen that the level of dispersion of the mass values in cases with additives is smaller. This may be because the additive plays a role in transforming wet friction to wet-rolling friction through making metal oxide elements present between the friction surfaces. In addition, it may also be a factor in helping the friction surfaces corrode more evenly.

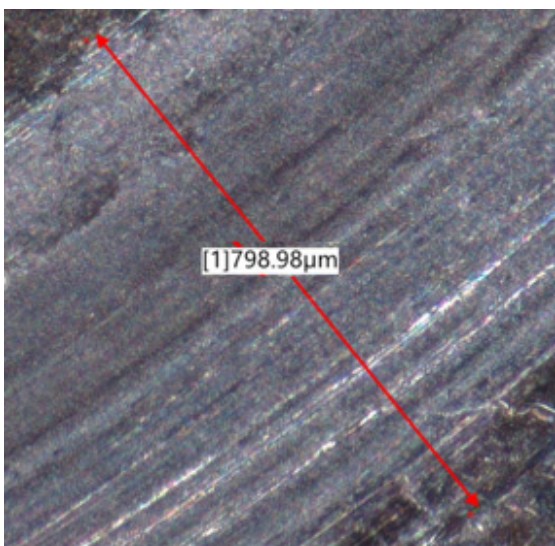

Scratch on upper ball, 0% additive

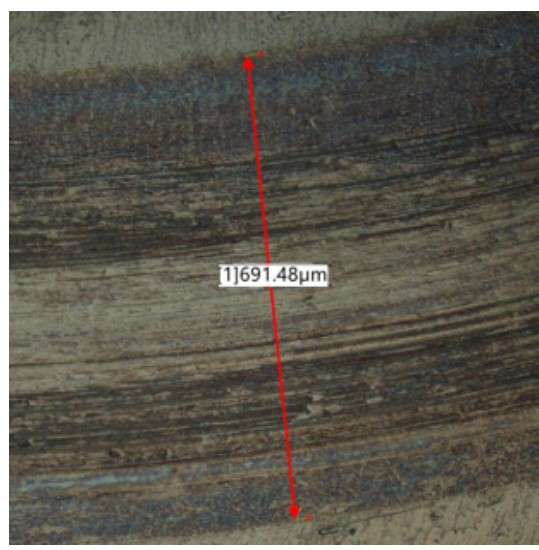

Scratch on upper ball, 0.5% additive

**Figure 3.** *Cont.*

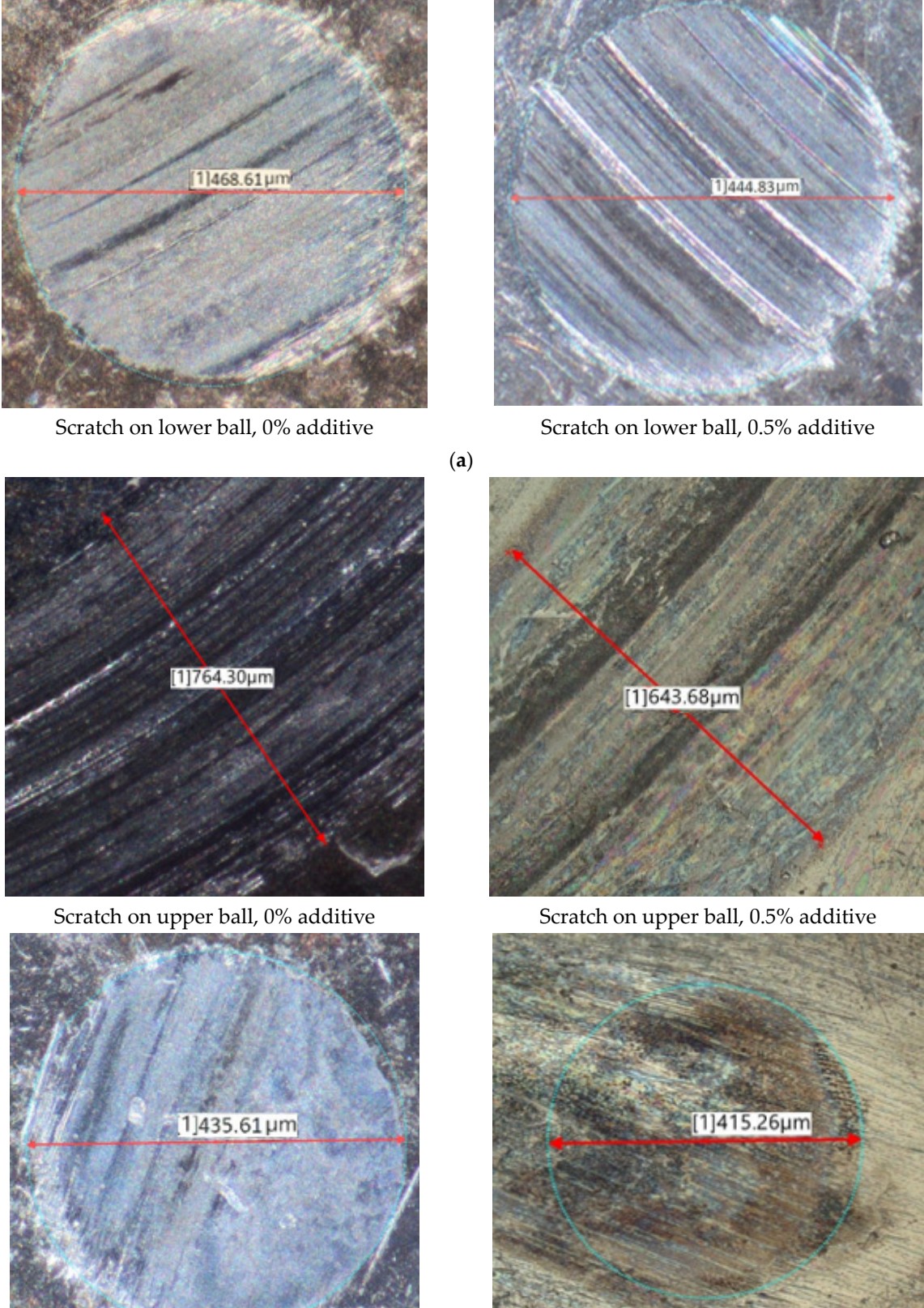

Scratch on lower ball, 0% additive　　　　　　　Scratch on lower ball, 0.5% additive

(**a**)

Scratch on upper ball, 0% additive　　　　　　　Scratch on upper ball, 0.5% additive

Scratch on lower ball, 0% additive　　　　　　　Scratch on lower ball, 0.5% additive

(**b**)

**Figure 3.** *Cont.*

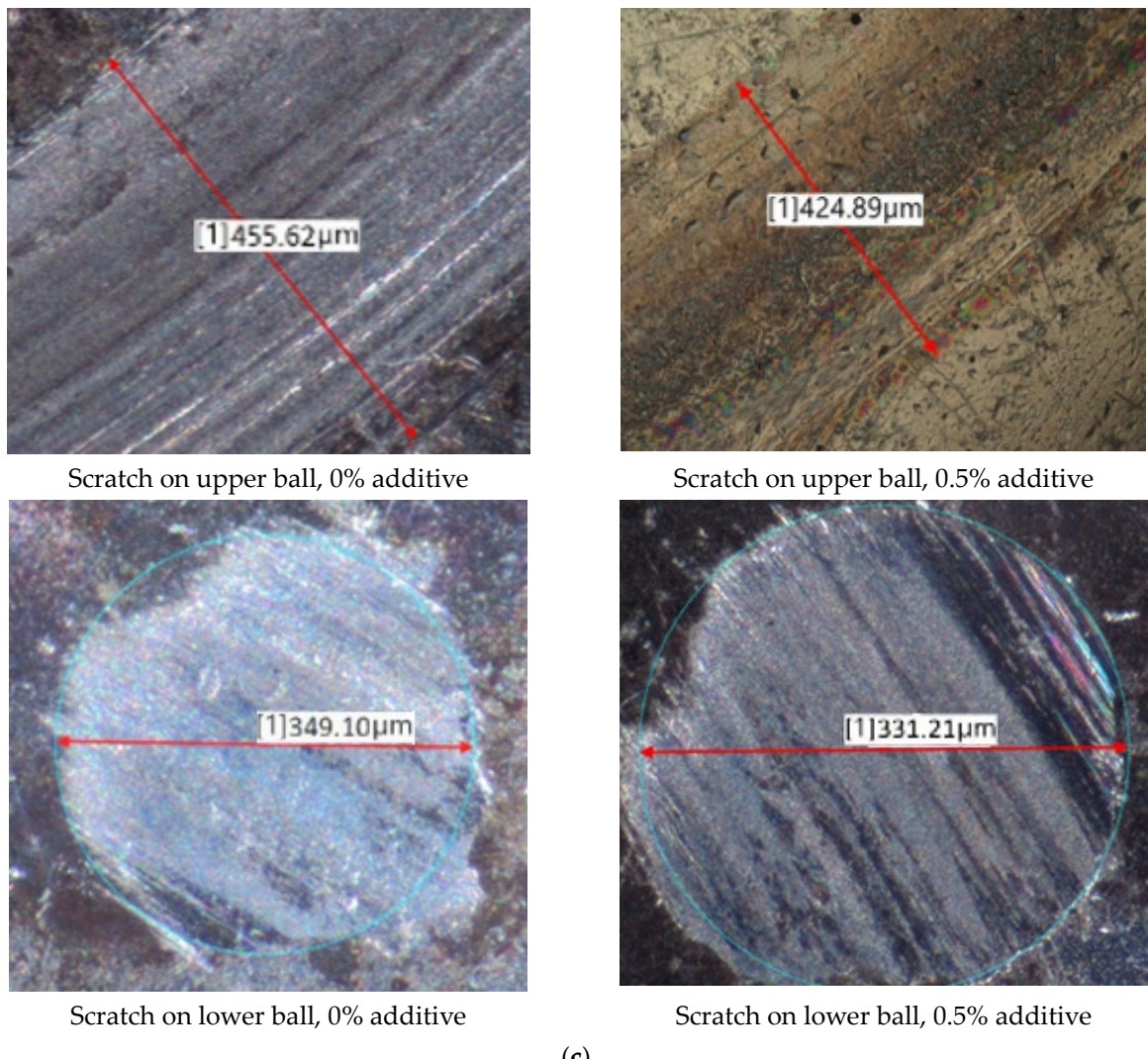

Scratch on upper ball, 0% additive

Scratch on upper ball, 0.5% additive

Scratch on lower ball, 0% additive

Scratch on lower ball, 0.5% additive

(**c**)

**Figure 3.** Surface images of balls experimented with industrial oils A (**a**), B (**b**), and C (**c**).

**Table 3.** Average mass of balls before and after we conducted experiments.

| Oils | 0% Fly Ash Additive | | | | 0.5% Fly Ash Additive | | | |
|---|---|---|---|---|---|---|---|---|
| | Before Experiments (g) | Standard Deviation | After Experiments (g) | Standard Deviation | Before Experiments (g) | Standard Deviation | After Experiments (g) | Standard Deviation |
| A | 8.346 | 0.0023 | 8.344 | 0.0018 | 8.345 | 0.0012 | 8.343 | 0.0009 |
| B | 8.328 | 0.0021 | 8.327 | 0.0019 | 8.336 | 0.0014 | 8.335 | 0.0014 |
| C | 8.329 | 0.0021 | 8.328 | 0.0021 | 8.325 | 0.0023 | 8.324 | 0.0021 |

In this study, the impact of additives or lubricants on the anti-wear properties of surface details was investigated through analyzing scratch marks on test balls according to ASTM standards. The scratch diameter of the balls was calculated as the average value and standard deviation, and these results are presented in Table 4 and Figure 5. Similarly, the scratch width of the top balls was measured, and the mean value and standard deviation were determined and reported in Table 5 and Figure 6. These results provide insight into the impact of additives or lubricants on the anti-wear properties of surface details, as assessed through examination of scratch marks on test balls using ASTM standards.

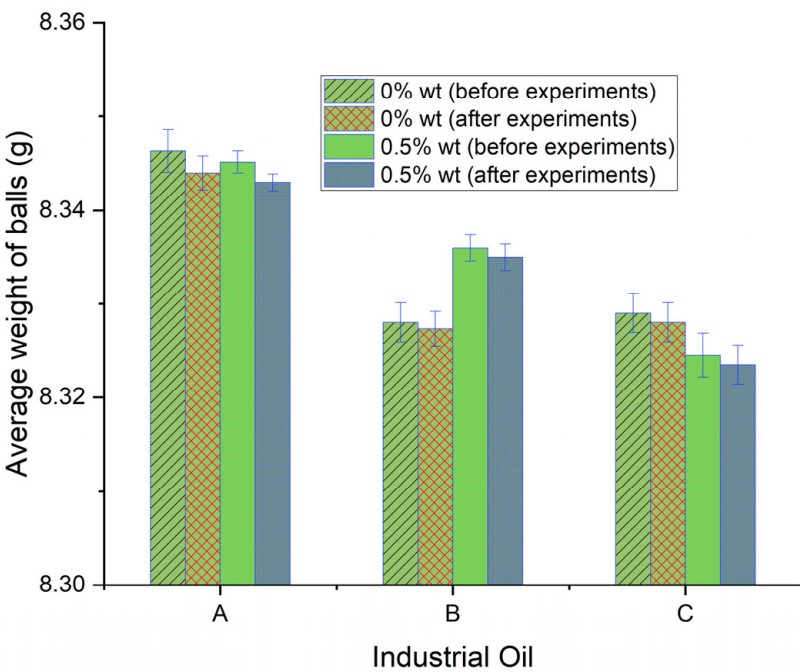

**Figure 4.** Average mass of balls after experiments with and without additive.

**Table 4.** Average scratch diameter of balls.

| Industrial Oils | 0% wt Fly Ash Additive | | 0.5% wt Fly Ash Additive | |
|---|---|---|---|---|
| | Average Scratch Diameter (mm) | Standard Deviation | Average Scratch Diameter (mm) | Standard Deviation |
| A | 0.461 | 0.014 | 0.455 | 0.018 |
| B | 0.426 | 0.017 | 0.417 | 0.004 |
| C | 0.338 | 0.016 | 0.333 | 0.012 |

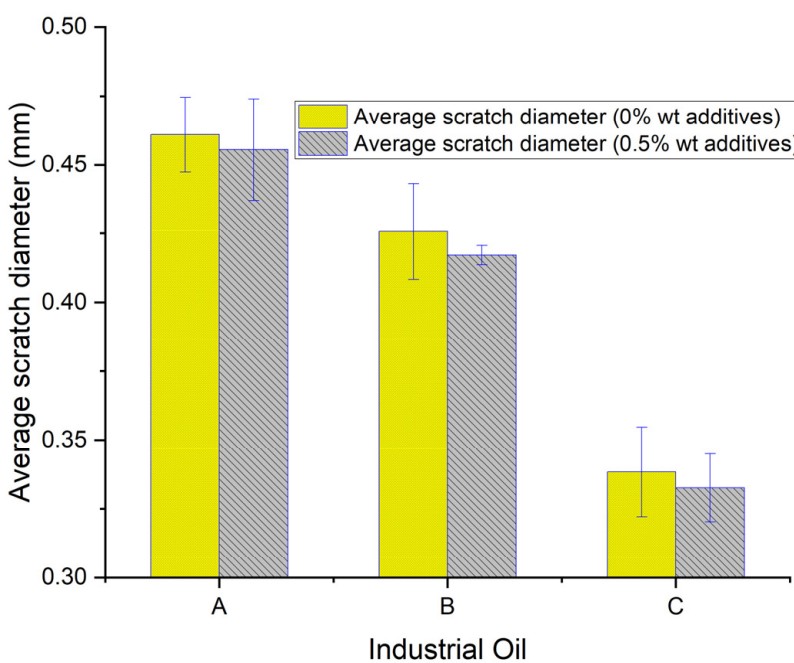

**Figure 5.** Effect of fly ash additive on the average diameter of ball scratches.

**Table 5.** Average scratch width of balls.

| Industrial Oils | 0% wt Fly Ash Additive | | 0.5% wt Fly Ash Additive | |
|---|---|---|---|---|
| | Average Scratch Width (mm) | Standard Deviation | Average Scratch Width (mm) | Standard Deviation |
| A | 0.784 | 0.014 | 0.676 | 0.013 |
| B | 0.753 | 0.011 | 0.643 | 0.011 |
| C | 0.461 | 0.012 | 0.434 | 0.009 |

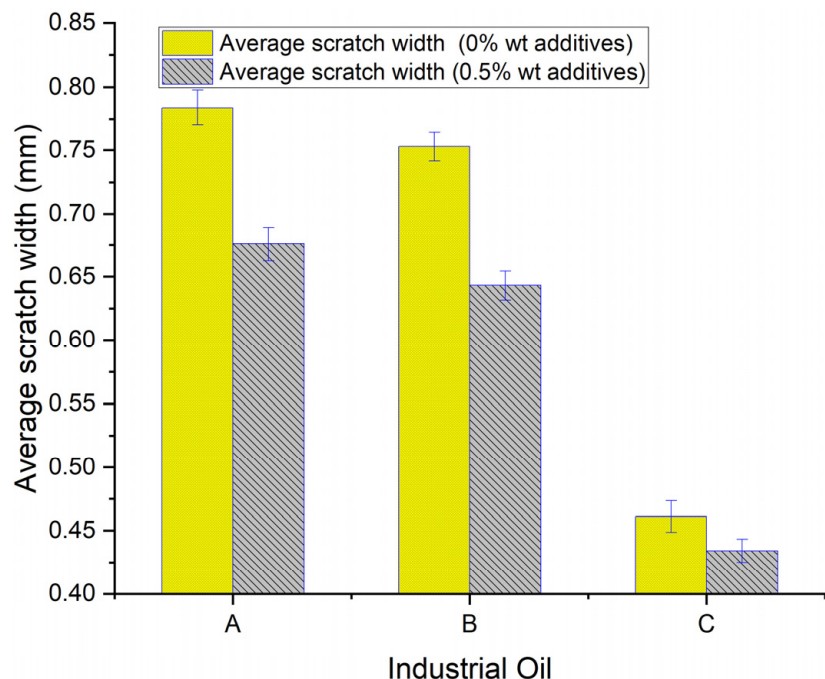

**Figure 6.** Effect of fly ash additive on average width of ball scratches.

Table 4 and Figure 5 demonstrate that oil C has the best wear resistance ability among the three types of oils tested, as it results in the smallest wear scar size. Conversely, oil A has the lowest wear resistance ability, as it has the largest wear scar size. The addition of 0.5% fly ash additive to the three types of oils resulted in a significant improvement in the wear resistance ability of oil B. The wear scar size decreased from 0.426 mm to 0.417 mm, and the standard deviation decreased from 0.017 to 0.004. This improvement was not as significant in oils A and C, with only small changes in their wear scar size. Oil A has a lower dynamic viscosity than oils B and C, allowing it to penetrate and lubricate detailed surfaces more easily. With the addition of fly ash additive, the viscosity of oils B and C increased, which reduced the wear scar size. However, the viscosity of oil A was initially low, and was not significantly affected by the fly ash additive.

These results indicate that adding 0.5% fly ash additive can improve the wear resistance ability of oil, though the improvement is dependent on the initial properties of each type of oil. The additive may also increase the viscosity of the oil and transform wet friction to wet-rolling friction through making metal oxide elements present between the friction surfaces, affecting its lubricating properties and wear resistance ability. The standard deviation in wear scar size also needs to be considered, as it indicates the uniformity of the measurement results, which is significant when adding the fly ash additive, indicating that the measurement results of oil B are more reliable after adding the additive.

Table 5 and Figure 6 present the scratch width and standard deviation of the balls lubricated with industrial oils A, B, and C, with and without the addition of 0.5% fly ash. The results demonstrate that the addition of fly ash has a significant impact on the anti-wear

properties of the oils, as evidenced by the scratch width of the balls in the friction test. Oil B outperforms oils A and C in terms of anti-wear properties, and this improvement is further evident with the addition of 0.5% fly ash, resulting in an increase of 14.6%, 13.8%, and 5.86%, respectively.

The scratch width for the ball bearing with oil C is narrower compared to oils A and B both with and without additive. The scratch width for the ball with oil A with additive is narrower than the scratch width for oil A without additive, and similar observations are made for oil B and C. The standard deviation for scratch width is small, indicating consistent and reliable results. Thus, the results suggest that the use of additives may have a positive effect on the performance of oils in reducing wear on mechanical components.

In summary, additives can improve oil viscosity, creating an oil film on machine components' surfaces. This film reduces wear and minimizes oil consumption during operation. However, excessively high viscosity may decrease lubrication efficiency and increase wear due to reduced oil ability. Optimal viscosity values may exist for each gap between two friction surfaces, suggesting that continuously increasing oil viscosity is not recommended and may lead to increased wear. The small metal oxide particles in the additive can transform sliding friction into rolling friction between machine components, resulting in lower friction and wear during operation. Further investigation will explore the effects of different additive ratios and high viscosities for each lubricant type to identify the most appropriate lubrication parameters.

## 4. Conclusions

The addition of fly ash as an additive to industrial oils is shown to improve their anti-wear properties. Oil B exhibited the best anti-wear performance among the oils tested, and the addition of fly ash significantly enhanced its performance. The scratch width of the balls in the four-ball wear test is a reliable indicator for evaluating oil anti-wear properties. It is worth noting that the concentration of fly ash additive in the oil mixture is critical in improving anti-wear properties. This result is because fly ash additive contains metal oxide elements that transform wet friction into wet-rolling friction, thereby improving the lubricating and wear-resistant properties of the oil. Based on the study's experiments, a concentration of around 0.5% based on weight of fly ash additive demonstrated a substantial improvement in anti-wear properties. This information is useful for manufacturers and researchers in selecting the appropriate concentration of fly ash in their oil formulations, enhancing anti-wear properties while maintaining the oil's physical and chemical characteristics. Thus, selecting the appropriate anti-wear oil is critical in ensuring machine component performance and service life: this study provides valuable information for choosing the optimal anti-wear oil for specific applications.

**Author Contributions:** Conceptualization, T.-A.B. and D.-T.N.; Project Administration, T.-A.B.; Supervision, T.-A.B. and D.-T.N.; Funding Acquisition: T.-A.B. and N.-T.B.; Methodology, T.-A.B. and V.-H.P.; Formal Analysis, T.-A.B. and V.-H.P.; Experimental data collection, N.-T.B., D.-T.N. and V.-H.P.; Writing—Review and Editing, T.-A.B., D.-T.N. and N.-T.B. All authors have read and agreed to the published version of the manuscript.

**Funding:** This research is funded by Vietnam Ministry of Education and Training under project number B2021-BKA-13.

**Institutional Review Board Statement:** Not applicable.

**Informed Consent Statement:** Not applicable.

**Data Availability Statement:** Not applicable.

**Acknowledgments:** The authors gratefully thank the financial support of Vietnam Ministry of Education and Training under Vietnam Grants B2021-BKA-13. The School of Mechanical Engineering and Hanoi University of Science and Technology are gratefully acknowledged for providing the support and conditions to carry out this research. This work was also supported by the Centennial

Shibaura Institute of Technology Action to mark the 100th anniversary of Shibaura Institute of Technology and entry into the top ten Asian Institutes of Technology list.

**Conflicts of Interest:** The authors declare no conflict of interest.

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
