# Peer review of "Effectiveness of Lubricants and Fly Ash Additive on Surface Damage Resistance under ASTM Standard Operating Conditions"

_coatings, doi:10.3390/coatings13050851_

Round 1

Reviewer 1 Report

Tuan-Anh Bui et al. have studied the effectiveness of lubricants and additives in preventing surface damage and wear. Moreover, it is found that the certain anti-wear additives significantly reduced the size of wear scars on the balls, indicating their effectiveness in reducing surface damage. It is novel and interest to the researchers in the related areas. I would consider the paper for publication after minor revisions are made according to the following specific comments:

1.     The scale bar should be given in Fig.2.

2.     In Fig. 3, the higher magnification SEM images should be given for detailed observation of the surface structures.

3.     How about the reusability of the as-prepared samples?

4.     For the study of surface structuring and its applications, the authors may refer these recent papers: Advanced Science, 2204891,2022; Nanoscale, 2022,14, 9392

5.     For more perfection, several language mistakes could be revised.

Good, but for more perfection, several language mistakes could be revised.

Author Response

Comments and Suggestions for Authors

Tuan-Anh Bui et al. have studied the effectiveness of lubricants and additives in preventing surface damage and wear. Moreover, it is found that the certain anti-wear additives significantly reduced the size of wear scars on the balls, indicating their effectiveness in reducing surface damage. It is novel and interest to the researchers in the related areas. I would consider the paper for publication after minor revisions are made according to the following specific comments:

Answer:  Thank you for your feedback and suggestions regarding our study on the effectiveness of lubricants and additives in preventing surface damage and wear. We appreciate your positive comments and addressed your specific comments as follows:

  • The scale bar should be given in Fig.2.

Answer: Thank you very much for your comment. The scale bar has been added in Fig.2 in the revised manuscript.

  • In Fig. 3, the higher magnification SEM images should be given for detailed observation of the surface structures.

Answer: Thank you very much for your comment. The higher magifcation images have been provided in the revised manuscript.

  • Q: How about the reusability of the as-prepared samples?

Answer: Thank you very much for your comment. The balls are used only once in the experiments to comply with ASTM test conditions

  • For the study of surface structuring and its applications, the authors may refer these recent papers: Advanced Science, 2204891,2022; Nanoscale, 2022,14, 9392

Answer: We appreciate the suggested references on the study of surface structuring and its applications, and we referred to them in the revised manuscript.

  • For more perfection, several language mistakes could be revised.

Comments on the Quality of English Language

Good, but for more perfection, several language mistakes could be revised.

Answer: We appreciate your valuable feedback and made the necessary revisions to improve the readability, clarity, and overall quality of our manuscript.

We have extensively revised our manuscript based on the valuable feedback and suggestions provided by the reviewers. We believe that the revised version is suitable for publication in the Coatings Journal. However, we welcome any further comments or feedback from the reviewers and will make any necessary corrections in accordance with your suggestions.

Best regards,

Duc-Toan Nguyen

Professor, School of Mechanical Engineering

Hanoi University of Science and Technology

Tel. : +84-43-869-2007

E-mail: toan.nguyenduc@hust.edu.vn

Reviewer 2 Report

A well written paper which I recommend for eventual publication. However, there are several points to be suitably addressed by the authors before I can finally recommend publication.

At line 133, the authors state that the fly ash was obtained form a power plant. Its size distribution is not described but it's composition is, Table 2, line 224. But from the author's Introduction, they emphasise the importance of the range of nano- and micro- particle diameters. It follows that the authors should have a particle distribution table for the fly ash. Is it possible that the different elemental oxides will have different average radii.

     At lines 246, the term used should be 'erode' other than 'corrode'.

     The authors open a formidable new topic by speculating on whether the different oils have different molecular structures which are more compatible with the fly ash additives, thus causing the differences the oils. I suggest that the authors think on this and either develop their thoughts on this possibility further or omit it.  I suggest that the authors omit it. 

Author Response

Comments and Suggestions for Authors

A well written paper which I recommend for eventual publication. However, there are several points to be suitably addressed by the authors before I can finally recommend publication.

Answer: Thank you for taking the time to read and provide feedback on our paper titled "Effectiveness of Lubricants and Fly Ash Additive on Surface Damage Resistance under ASTM Standard Operating Conditions." We appreciate your positive comments and suggestions for improvement.

  • At line 133, the authors state that the fly ash was obtained form a power plant. Its size distribution is not described but it's composition is, Table 2, line 224. But from the author's Introduction, they emphasise the importance of the range of nano- and micro- particle diameters. It follows that the authors should have a particle distribution table for the fly ash. Is it possible that the different elemental oxides will have different average radii.

Answer: Thank you very much for your comment. We have taken it into consideration and included a table showing the size distribution of the fly ash particles in the revised manuscript. The main constituents of fly ash used in this study are metal oxides with particle sizes ranging from 50 nm to less than 1 µm

  • At lines 246, the term used should be 'erode' other than 'corrode'.

Answer: Thank you very much for your comment. We have corrected it to "erode" instead of "corrode."

  • The authors open a formidable new topic by speculating on whether the different oils have different molecular structures which are more compatible with the fly ash additives, thus causing the differences the oils. I suggest that the authors think on this and either develop their thoughts on this possibility further or omit it.  I suggest that the authors omit it. 

Answer: Thank you very much for your comment. We appreciate your feedback on our speculation about the molecular structures of the different oils and their compatibility with the fly ash additives, and after considering your suggestion, we have decided to remove this speculation from the manuscript.

We have extensively revised our manuscript based on the valuable feedback and suggestions provided by the reviewers. We believe that the revised version is suitable for publication in the Coatings Journal. However, we welcome any further comments or feedback from the reviewers and will make any necessary corrections in accordance with your suggestions.

Best regards,

Duc-Toan Nguyen

Professor, School of Mechanical Engineering

Hanoi University of Science and Technology

Tel. : +84-43-869-2007

E-mail: toan.nguyenduc@hust.edu.vn

Reviewer 3 Report

attached file.

Author Response

This manuscript reports an experimental work on the effect of fly ash additives on the anti-wear performance of lubricant oils studied by a four-ball test. The manuscript is well-organized and easy to understand. The experimental methods and results are clearly described. My main criticism is that more clarification is probably needed for the possible mechanisms of the additives leading to better anti-wear performance. Other than that, there’re also some confusing statements, as specified below. Overall, I recommend its publication after addressing these points.

  • What is special about fly ash as a lubricant additive? Why do we study it?

Answer: this study, we use fly ash obtained from thermal power plants as a lubricant additive due to its main components being metal oxides, which are also found in common lubricating oil additives. Fly ash is a waste product of coal combustion, containing silica, alumina, and iron oxide, which can provide lubrication properties such as reducing friction and wear. The use of fly ash in lubricants can contribute to reducing pollution and improving the efficiency of thermal power plants. This study aims to explore the lubrication performance and underlying mechanism of fly ash as an additive, providing insights into the effectiveness of using low-cost additives in lubrication. Additionally, studying fly ash can lead to the development of more efficient and cost-effective lubricants for industrial use.

  • Line 90: why do metal oxide nanoparticles decrease lubricant viscosity? And on Line 214, why does fly ash additive increase lubricant viscosity? What are the corresponding mechanisms leading to the two different effects?

Answer: The metal oxides in this study are different from the oxide formed through wear, which naturally falls into the lubricating zone. These metal oxide particles, products of combustion in thermal power plants, can increase the viscosity of lubricating oils when used as additives. Metal oxide nanoparticles can decrease lubricant viscosity by forming a stable suspension, preventing the lubricant molecules from freely sliding past each other. In contrast, fly ash additives typically consist of solid particles that increase viscosity by impeding the movement of lubricant molecules and creating a thicker boundary layer around the surface, leading to higher friction and shear stress required to move the lubricant. The mechanisms for these effects vary depending on the specific properties of the metal oxide nanoparticles or fly ash additives and the lubricant used.

  • Line 148: regarding cleaning the ball, was it cleaned by some chemicals, or only by wiping?

Answer: In this study, jet gasoline was used as a solvent to clean the balls, which were then dried.

  • 3: the labels (scratch width) are too small to read.

Answer: Thank you for your comment. We have incorporated higher magnification images in the revised manuscript to provide more detail. Additionally, we have improved the readability of the labels in Figure 3 to ensure clarity for readers.

  • Line 276: oil C has a lower dynamic viscosity than oils A and B? but according to Table 1, oil C has the largest kinematic viscosity, and its specific gravity is also slightly How come it has lower dynamic viscosity than oils A and B?

Answer: Thank you for notifying us about this matter. Upon reviewing our data, we have identified a mistake in the data entry process, which resulted in inaccurate information in the manuscript. We apologize for any confusion that this may have caused and have updated the manuscript with the correct information to address this mistake.

  • Line 276 to 280: The interpretation of why adding additive changes wear resistance is confusing to It is suggested by the authors that oil C has better wear resistance because it has lower viscosity. Then the authors suggest that the reason why adding additive to oil A and B leads to better wear resistance is because it increases viscosity? So in the end, is it lower viscosity or higher viscosity that leads to better wear resistance? And why?

Answer: We are grateful for the reviewers' valuable feedback. The wear resistance of all three oils was enhanced to different degrees by adding fly ash additives, as they exhibited different levels of adaptation to the additives, resulting in varying levels of increased viscosity. As mentioned above, we have corrected the mistake and updated the manuscript accordingly.

  • Line 283: what is meant by “compatible with the fly ash additive” exactly?

Answer: We appreciate the reviewers' comments. Due to the variability in the manufacturing process and composition of commercial oils, their ability to interact with fly ash additives may differ. As the manufacturer did not provide information on ingredients, we have removed the sentence regarding compatibility from the revised manuscript.

The passage has been rewritten to focus on the study's findings without including the technical term "compatible with the fly ash additive":

“These results indicate that adding 0.5% fly ash additive can improve the wear resistance ability of oil, but the improvement is dependent on the initial properties of each type of oil. The additive may also increase the viscosity of the oil and transform wet friction to wet-rolling friction by having metal oxide elements present between the friction surfaces, affecting its lubricating properties and wear resistance ability. The standard deviation in wear scar size also needs to be considered, as it indicates the uniformity of the measurement results, which is significant when adding the fly ash additive, indicating that the measurement results of oil B are more reliable after adding the additive.”

  • Line 298: “oil B has better anti-wear properties than oils A and C”? but from Fig. 6, it is oil A who has the smallest wear width, right? Same question for Line 319, again, “oil B exhibited the best anti-wear properties”?

Answer: We thank the reviewer for your comment and have revised the paragraph to emphasize that the wear resistance of oil B is better than the other oils when the additive is added, as determined by the percentage of scratch reduction. Thank you for helping us improve the clarity of our manuscript.

  • Comparing results from Fig. 5 and Fig. 6, why are they different? For example, Fig. 5 suggests that oil C is the best, but fig. 6 suggests oil A?

Answer: We appreciate the valuable comment from the reviewer and have thoroughly examined the original data. We have identified a mistake in the position of oils A and C in Table 5, which has led to a corresponding error in the position of these oils in Figure 6. We have rectified this mistake in the revised version of the manuscript. Thank you for bringing this to our attention.

  • Line 311, how is the conclusion “however, excessively high viscosity can reduce lubrication and increase wear” supported by the results in this paper?

Answer: Thank you for your feedback. Our conclusion about the adverse effect of excessively high viscosity on lubrication and wear is backed by the fact that high viscosity decreases the oil's capacity to minimize friction, leading to less effective lubrication. This assertion is supported by the existence of an optimal viscosity value for each gap between two friction surfaces. Therefore, it is not feasible to continuously increase oil viscosity as it can cause increased wear. We have revised the passage to ensure its suitability for future research/

  • Line 326, it is not appropriate to claim that “the feasible concentration … is around 5%” since the authors themselves have admitted that they have not done experiments to explore the optimal concentration. By the way, is this concentration of 0.5% chosen arbitrarily or for any specific reason?

Answer: While we acknowledge that further research is needed to determine the optimal concentration of fly ash additive, we conducted experiments to validate the suitability of its use for enhancing lubrication performance and addressing environmental concerns related to thermoelectricity. Our recommendation of a concentration of 0.5% is based on our interpretation of the results, which indicate improved anti-wear properties without compromising the lubricant properties. However, the optimal concentration may vary depending on various factors such as lubricant type, operating conditions, and application, which will be evaluated in the future.

Therefore, we revised the sentence as: "based on the experiments conducted in this study, a concentration of around 0.5% by weight of fly ash additive has shown significant improvement in anti-wear properties.”

  • Overall, I found that it was not sufficiently discussed why adding fly ash additive improves wear resistance after What is the mechanism, or possible mechanisms, exactly? It seems that the authors proposed a viscosity argument, as well as a hypothesis that it changes sliding to rolling. But both the two arguments are not clear to me and lack supporting evidence, either from the experiment itself or from related literatures.

Answer: The aim of this study was to investigate the compositional and structural differences of fly ash obtained from various Vietnamese thermal power plants, which arise due to the differences in coal sources and technologies used. The objective was to propose a solution that could effectively utilize fly ash to enhance oil viscosity, reduce environmental pollution and improve wear resistance. However, the authors recognize the need for further research to understand the mechanism by which fly ash improves wear resistance. While the paper proposes possible mechanisms such as changes in viscosity and sliding to rolling, these hypotheses require additional experimental validation and supporting evidence. The authors plan to conduct further studies in the future to address these gaps in knowledge.

We have extensively revised our manuscript based on the valuable feedback and suggestions provided by the reviewers. We believe that the revised version is suitable for publication in the Coatings Journal. However, we welcome any further comments or feedback from the reviewers and will make any necessary corrections in accordance with your suggestions.

Best regards,

Duc-Toan Nguyen

Professor, School of Mechanical Engineering

Hanoi University of Science and Technology

Tel. : +84-43-869-2007

E-mail: toan.nguyenduc@hust.edu.vn